# Metabolomic Biomarkers of Dietary Approaches to Stop Hypertension (DASH) Dietary Patterns in Pregnant Women

**DOI:** 10.3390/nu16040492

**Published:** 2024-02-08

**Authors:** Liwei Chen, Jin Dai, Guoqi Yu, Wei Wei Pang, Mohammad L. Rahman, Xinyue Liu, Oliver Fiehn, Claire Guivarch, Zhen Chen, Cuilin Zhang

**Affiliations:** 1Department of Epidemiology, Fielding School of Public Health, University of California, Los Angeles, CA 90095, USA; cliwei86@g.ucla.edu (L.C.); daijin@g.ucla.edu (J.D.); xinyue2396@g.ucla.edu (X.L.); 2Global Center for Asian Women’s Health, Yong Loo Lin School of Medicine, National University of Singapore, Singapore 117597, Singapore; yu_04@nus.edu.sg (G.Y.); obgpww@nus.edu.sg (W.W.P.); guivarch@nus.edu.sg (C.G.); 3Department of Obstetrics and Gynaecology, Yong Loo Lin School of Medicine, National University of Singapore, Singapore 117597, Singapore; 4Bia-Echo Asia Centre for Reproductive Longevity and Equality (ACRLE), Yong Loo Lin School of Medicine, National University of Singapore, Singapore 117597, Singapore; 5Division of Cancer Epidemiology and Genetics, National Cancer Institute, National Institutes of Health, Rockville, MD 20850, USA; mohammad.rahman2@nih.gov; 6West Coast Metabolomics Center, UC Davis Genome Center, University of California, Davis, CA 95616, USA; ofiehn@ucdavis.edu; 7Division of Population Health Research, Eunice Kennedy Shriver National Institute of Child Health and Human Development, National Institute of Health, Bethesda, MD 20892, USA; chenzhe@mail.nih.gov

**Keywords:** DASH diet, biomarkers, metabolomics, pregnant women

## Abstract

**Objective:** the aim of this study was to identify plasma metabolomic markers of Dietary Approaches to Stop Hypertension (DASH) dietary patterns in pregnant women. **Methods:** This study included 186 women who had both dietary intake and metabolome measured from a nested case-control study within the NICHD Fetal Growth Studies–Singletons cohort (FGS). Dietary intakes were ascertained at 8–13 gestational weeks (GW) using the Food Frequency Questionnaire (FFQ) and DASH scores were calculated based on eight food and nutrient components. Fasting plasma samples were collected at 15–26 GW and untargeted metabolomic profiling was performed. Multivariable linear regression models were used to examine the association of individual metabolites with the DASH score. Least absolute shrinkage and selection operator (LASSO) regression was used to select a panel of metabolites jointly associated with the DASH score. **Results:** Of the total 460 known metabolites, 92 were individually associated with DASH score in linear regressions, 25 were selected as a panel by LASSO regressions, and 18 were identified by both methods. Among the top 18 metabolites, there were 11 lipids and lipid-like molecules (i.e., TG (49:1), TG (52:2), PC (31:0), PC (35:3), PC (36:4) C, PC (36:5) B, PC (38:4) B, PC (42:6), SM (d32:0), gamma-tocopherol, and dodecanoic acid), 5 organic acids and derivatives (i.e., asparagine, beta-alanine, glycine, taurine, and hydroxycarbamate), 1 organic oxygen compound (i.e., xylitol), and 1 organoheterocyclic compound (i.e., maleimide). **Conclusions:** our study identified plasma metabolomic markers for DASH dietary patterns in pregnant women, with most of being lipids and lipid-like molecules.

## 1. Introduction

The DASH (Dietary Approaches to Stop Hypertension) dietary pattern was originally developed to reduce blood pressure [1]. Recently, growing evidence suggests that adherence to the DASH dietary pattern has beneficial effects on type 2 diabetes (T2D), cardiovascular diseases (CVD), metabolic syndrome, and Alzheimer’s disease, in addition to hypertension [2,3,4,5]. A recent randomized cross-over study also showed that the DASH diet significantly reduced the circulation levels of serum high-sensitivity C-reactive protein (CRP: a biomarker of chronic inflammation) among children, independent of changes in body weight and lipid profiles [6]. In pregnant women, adherence to the DASH dietary pattern has also been linked to a lower risk of pre-term birth [7,8]. Therefore, DASH has been recommended by the American Dietary Guidelines as a healthy dietary pattern for Americans. However, the underlying molecular mechanisms which explain the health benefits of the DASH dietary pattern have yet to be elucidated.

Biomarkers of dietary intake can be applied as objective measures of dietary patterns and can help to reveal the underlying molecular mechanisms between dietary patterns and health outcomes [9]. Recent advances in high-throughput metabolomic profiling techniques have paved a new and comprehensive approach to discovering the biomarkers of food intake and patterns [10]. Metabolomic markers of dietary patterns may collectively represent an intermediate phenotype of the underlying metabolic state and reflect the intersection of genetics, diet, and metabolism. Pioneer work using the metabolomic approach has identified promising metabolites as markers for dietary patterns, including DASH, in the general population [11,12,13]. Unfortunately, such investigations in pregnant women are still lacking. Therefore, we aimed to identify metabolomic markers of DASH dietary patterns in pregnant women.

## 2. Methods

### 2.1. Study Population and Design

This was a prospective study among women who were enrolled in the Eunice Kennedy Shriver National Institute of Child Health and Human Development (NICHD) Fetal Growth Studies–Singletons cohort (FGS). Between 2009 and 2013, a total of 2802 women aged 18–40 years with singletons were recruited between 8 and 13 gestational weeks (GW) at 12 clinical sites in the United States (US). The current study only included women with data on both blood plasma metabolomics and dietary intakes from a nested gestational diabetes mellitus (GDM) case-control study within the FGS. Details of the cohort have been described previously [14]. Institutional Review Board approval was obtained (IRB number: 09-CH-N152) in May 2009 for all participating clinical sites, the data coordinating center, and NICHD. All participants provided informed consent prior to data collection. The FGS is registered at the ClinicalTrials.gove (Identifier: NCT00912132).

### 2.2. Assessment of Dietary Intakes and DASH Score

The women’s dietary intake in the past 3 months was assessed at study enrollment (8–13 GW), using a Food Frequency Questionnaire (FFQ) which was developed and validated by the National Cancer Institute [15,16,17]. Adherence to the DASH dietary pattern was assessed by the DASH score, calculated based on the consumption of 8 food and nutrient components (i.e., vegetables, whole fruits, whole grains, nuts and legumes, low-fat dairy, red and processed meats, sweetened beverages, and sodium) using a published method [18]. A higher DASH score indicates a better adherence to DASH dietary patterns.

### 2.3. Biospecimen Collection, Metabolomic Profiling, and Data Pre-Treatment

Fasting blood samples were collected at 15–26 GW and were immediately processed. Plasma specimen were stored at −80 °C until analysis. Untargeted metabolomic profiling was performed at the West Coast Metabolomics Center, University of California Davis (UC-Davis), using liquid chromatography–high-resolution mass spectrometry (LC-MS) for lipidomics [19] and gas chromatography–mass spectrometry (GC-MS) for primary metabolomics [20]. Internal standards were applied for the calibration of retention times. Among a total of 751 detected features, 460 were annotated metabolites and 291 were unknown. In the current study, we only included the 460 annotated metabolites for a better interpretation and comparison to other studies. All 460 metabolites had missing values <20% and missing values were imputed to half of the minimum value by batch and visits [21]. Metabolites were re-scaled to a median of 1 and then log-transformed to correct the day-to-day variation from the platform [12].

### 2.4. Covariates

The women’s sociodemographic characteristics as well as reproductive and medical history were collected at study enrollment using a detailed questionnaire. Pre-pregnancy body weights were self-reported and heights were directly measured at study enrollment. Pre-pregnancy body mass index (BMI) was calculated by dividing weight (kg) by square of height (m). Women were classified as normal weight (<25.0 kg/m^2^), overweight (25.0–29.9 kg/m^2^), or obese (≥30.0 kg/m^2^) by their pre-pregnancy BMI. Maternal physical activity in the past 12 months was assessed using a validated Pregnancy Physical Activity Questionnaire (PPAQ) [22] at study enrollment.

### 2.5. Statistical Methods

Sampling weights were calculated by a statistician and applied to all of the statistical analyses to represent the full NICHD Fetal Growth Studies–Singletons population. The women’s characteristics at the study enrollment were compared across the tertiles of DASH score. Data were presented as weighted percentage (%) and actual frequency (N) for categorical variables and weighted mean (standard errors, SE) for continuous variables. *p*-values comparing women across DASH score tertiles were obtained by one-way ANOVA tests for continuous variables and χ^2^ tests for categorical variables.

Several steps were performed to identify plasma metabolites that are associated with DASH scores. First, multivariable linear regression models were utilized to estimate the associations between DASH score and individual fasting plasma metabolite (dependent variable), adjusting for maternal age (years), race (non-Hispanic White, non-Hispanic Black, Hispanic, and Asian and Pacific Islander), education (high-school degree or less, associated degree, and bachelor’s degree or more), pre-pregnancy BMI (kg/m^2^), and PA (minutes per week). Second, a panel of plasma metabolites jointly associated with higher DASH scores (tertile 3 vs. tertile 1) were selected from the 460 known metabolites using the least absolute shrinkage and selection operator (LASSO) regression models with 10-fold cross-validation. Joint associations were validated using the high dimensional inference model (B = 300), adjusting for the covariates [23,24]. For this analysis, participants were randomly partitioned into training and validation sets at a 2:1 ratio. The predictive capacity of the metabolites that were jointly associated with DASH scores was assessed through the area under the curve (AUC). Lastly, we examined the associations of metabolites (dependent variable) with each DASH food component using multivariable linear regression models adjusted for the same set of covariates as above. *p*-values after the Benjamini–Hochberg correction with the false discovery rate (FDR) <0.05 were considered as statistically significant [25]. The data analyses were conducted using SAS software (version 9.4; SAS Institute, Cary, NC, USA) and R (version 4.0.2; R Studio: Integrated Development for R. R Studio, Inc., Boston, MA, USA).

## 3. Results

### 3.1. Women’s Baseline Characteristics

Among all women, 27.2% of them were non-Hispanic Whites, 25.7% were non-Hispanic Blacks, 24.4% were Hispanics, and 22.7% were Asian and Pacific Islanders. The mean (SE) age of women at enrollment was 28.00 years (0.40). Compared to women in the lowest tertile of DASH score (T1), women in the highest tertile (T3) were more likely to be non-Hispanic Whites and nulliparous, had a higher level of education, and had private health insurance or managed care, but were less likely to be married or live with a partner, or to born in the US; they also had lower pre-pregnancy BMI, but higher intakes of total energy, carbohydrate, protein, MUFA, PUFA, dietary fiber, vegetables, fruits, whole grain, nuts and legumes, and low-fat dairy, and lower intakes of sugar-sweetened beverages (Table 1).

### 3.2. Associations of Dietary DASH Score with Plasma Metabolites

Of the total 460 known metabolites, 345 were lipids or lipid-like molecules (75%), followed by 55 organic acids and derivatives (12.0%), 30 organic oxygen compounds (6.5%), 12 organoheterocyclic compounds (2.6%), 10 organic nitrogen compounds (2.2%), and 8 “others” (including 3 homogeneous non-metal compounds, 3 nucleosides, nucleotides, and analogues, and 2 benzenoids).

After adjusting for age, race, education, pre-pregnancy BMI, and physical activity, 92 metabolites were individually associated with DASH score with the FDR <0.05 (Appendix A) in linear regression. Among the 92 metabolites, 78 (84.8%) were lipids and lipid-like molecules, followed by 7 (7.6%) organic acids and derivatives, 3 (3.3%) organic oxygen compounds, 2 (2.2%) organoheterocyclic compounds, 1 (1.1%) benzenoid, and 1 (1.1%) homogeneous non-metal compound (Appendix A). A panel of 25 metabolites were jointly associated with DASH score in the LASSO regression with an AUC of 0.79 (95% CI, 0.65–0.92) (Figure 1). Overall, 18 metabolites (an AUC of 0.73 (95% CI, 0.56–0.90)) were selected by both linear and LASSO regressions (Table 2), with 14 of them having positive associations and 4 having inverse associations. The majority (11 of 18) of the DASH-related metabolites were lipids and lipid-like molecules (i.e., TG (49:1), TG (52:2), PC (31:0), PC (35:3), PC (36:4) C, PC (36:5) B, PC (38:4) B, PC (42:6), SM (d32:0), γ-tocopherol, and dodecanoic acid). The rest were five organic acids and derivatives (i.e., asparagine, β-alanine, glycine, taurine, and hydroxycarbamate), one organic oxygen compound (i.e., xylitol), and one organoheterocyclic compound (i.e., maleimide). Of the 18 DASH-related metabolites, 10 were associated with low-fat dairy (i.e., TG (49:1), PC (31:0), PC (35:3), PC (38:4) B, PC (42:6), SM (d32:0), dodecanoic acid, beta-alanine, glycine, and hydroxycarbamate), 5 with sodium (i.e., PC (31:0), PC (35:3), PC (38:4) B, SM (d32:0), and asparagine), 2 with fruits (i.e., PC (36:4) C and dodecanoic acid), and 1 with whole grains (i.e., PC (31:0)) (Table 2).

## 4. Discussion

In this prospective cohort study among racially diverse pregnant women, we identified plasma metabolomic biomarkers for DASH dietary patterns. The top 18 metabolites which were both individually and jointly associated with DASH scores included 11 lipids and lipid-like molecules (i.e., TG (49:1); TG (52:2), PC (31:0), PC (35:3), PC (36:4) C, PC (36:5) B, PC (38:4) B, PC (42:6), SM (d32:0), gamma-tocopherol, and dodecanoic acid), 5 organic acids and derivatives (i.e., asparagine, beta-alanine, glycine, taurine, and hydroxycarbamate), 1 organic oxygen compound (i.e., xylitol), and 1 organoheterocyclic compound (i.e., maleimide). Most of these DASH-related metabolomic marks (10 of 18) were associated with dairy consumption.

To date, the most rigorous investigations that identified DASH-related metabolomic markers were studies utilizing data from randomized controlled feeding studies, including the DASH trial [12] and DASH Mechanism Study [26]. Several observational studies also identified circulating metabolomic markers for DASH dietary patterns in general populations, including the Cancer Prevention Study (CPS)-II Nutrition Cohort among postmenopausal women [27], the Framingham Offspring Study [28], the Atherosclerosis Risk in Communities Study (ARIC) [29], the Chronic Renal Insufficiency Cohort (CRIC) study [30], and the Insulin Resistance Atherosclerosis Study (IRAS) [31]. Although most DASH-related metabolites are lipids and lipid-like molecules, among these studies conducted with general populations, only a few metabolites were replicated. Of the 18 top metabolites selected in our study among pregnant women, 3 metabolites (e.g., SM (d32:0), TG (52:2), and gamma-tocopherol) were also reported in other studies conducted with general adult populations [12,26,28,29] and postmenopausal women [32]. It is noteworthy that work in this field is still in its early stage and different studies have applied various methods in metabolomic profiling, signal processing, and data analysis. The variable metabolites identified from different studies may reflect the variability in the physiological conditions of study populations (e.g., pregnant vs. non-pregnant) or blood samples used in different studies (fasting vs. non-fasting blood), but is also likely due to the methods in metabolomic profiling (targeted vs. untargeted, or mass spectrometry vs. nuclear magnetic resonance spectroscopy), or the demographic differences in the populations.

From the literature, the most consistent metabolite that has been related to the DASH dietary pattern is γ-tocopherol, which was inversely associated with the DASH dietary pattern in our study and four previous studies (including the DASH trial) [12,26,29,32], suggesting γ-tocopherol could be an important biomarker for the DASH diet. Although γ-tocopherol is the most abundant isoform of vitamin E (i.e., α-tocopherol, γ-tocopherol, β-tocopherol, and δ-tocopherol) in the American diet, α-tocopherol is the dominant isoform in human circulation and contributes to the antioxidant properties of vitamin E [33,34]. Emerging evidence from both human and animal studies suggests that γ-tocopherol and α-tocopherol have opposing regulatory functions during inflammation [35,36]. In human observational studies, a higher serum level of γ-tocopherol was associated with impaired lung function, higher levels of fasting glucose, and increased risk of hemorrhagic stroke mortality, whereas a higher serum level of α-tocopherol had the opposite associations [37,38,39]. Thus, a part of the DASH dietary pattern’s health benefits may be related to its effect on lowering the circulation level of γ-tocopherol.

Several amino acid metabolites (i.e., asparagine, beta-alanine, glycine, and taurine) were selected as biomarkers and were positively associated with the DASH diet in our study. These amino acid derivatives are mainly involved in protein biosynthesis. Asparagine is a non-essential α-amino acid that is metabolized from aspartic acid. It is widespread in many foods, but high amounts can be found in dairy products, eggs, fish, seafood, beef, and poultry. In humans, asparagine’s primary role is for protein biosynthesis but it is also implicated in cancer cell proliferation [40] and vessel formation [41]. β-alanine is the only naturally occurring β-amino acid and is particularly rich in meat products such as chicken, beef, pork, and fish. β-alanine has been identified as the biomarker of red meat consumption in a previous study [42] but was related to low dairy consumption in our study. In humans, β-alanine is a critical precursor of carnosine (a major contributor to H^+^ buffering during high-intensity exercise) biosynthesis, and thus, it is a popular ergogenic supplement for sports performance [43]. There is also a growing number of investigations on the effect of exogenous carnosine and β-alanine on myocardial health, but mainly using animal models [44]. The clinical studies in human participants supplemented with β-alanine or carnosine are still lacking. Glycine can be found in a wide range of foods, including fruits, vegetables, legumes, fish, dairy products, and meat. In humans, glycine is one of twenty proteinogenic amino acids of which the primary function is for protein biosynthesis, but it also plays a role as a neurotransmitter in the central nervous system and holds antioxidant, anti-inflammatory, and immunomodulatory functions in the peripheral and nervous tissues [45]. A previous study found that plasma glycine level was low in individuals with obesity or diabetes [46]. Taurine is a sulfur amino acid that can be biosynthesized from cysteine in humans but with limited biosynthetic ability. Thus, circulating taurine is partially obtained from foods such as shellfish and the dark meat of turkey and chicken [47]. Taurine has broad biological functions including serving as a neurotransmitter in the brain, a stabilizer of cell membranes, and a facilitator in the transport of ions (i.e., sodium, potassium, calcium, and magnesium). Taurine has been linked to coronary artery disease, blood pressure, plasma cholesterol, and myocardial function in animal models, but the evidence from epidemiologic studies is limited [48]. Taken together, the positive associations of these amino acids and their derivatives with the DASH diet suggest their potential roles in the underlying mechanisms and warrant further investigations.

A few metabolites belonging to the pyrrolidines and carbohydrate classes were also selected as metabolomic markers for the DASH dietary pattern, but our knowledge about these metabolites is limited. Maleimide is not a naturally occurring metabolite and is only found in those individuals exposed to this compound or its derivatives [49]. Xylitol is a sugar alcohol that can be found naturally in many fruits and vegetables, or as a sugar substitute in sugar-free products. Thus, the positive associations of xylitol with the DASH diet may reflect the high consumption of fruit and vegetables, or some commercially labeled “sugar-free” products [50].

We found that most of the metabolites associated with the DASH diet among pregnant individuals were lipids or lipid-like species; this is consistent with previous studies and also with findings from the DASH trial. The majority of the DASH diet-related lipid metabolites belong to long-chain TGs and PCs and are associated mainly with the consumption of dairy products, fruit, and nuts, all three of which are important components of the DASH diet. TGs and PCs with different acyl chains and double bonds have different biological functions, but the exact health effects of each or the combination of these lipid metabolites are still poorly understood [51,52]. Sphingomyelins (SMs) belong to the sphingolipids superclass. They are important components of cell membranes and play key roles in signal transduction in humans. They are found in dairy products but can also be synthesized endogenously [53]. Sphingomyelin (32:0) was identified as a biomarker for the DASH dietary pattern in our study and in the DASH trial [12].

Our study has several unique strengths. First, our participants were recruited from 12 health centers in the US and included multiple racial groups, thus increasing the generalizability of the study findings to US pregnant women. Second, we calculated the DASH score using a predefined method which can be easily reproduced in other studies. Third, we collected fasting plasma samples, which are subjected to less measurement variability. Furthermore, we performed multivariate methods to identify panels of DASH-related metabolites. As dietary patterns are combinations of foods and nutrients, a panel of metabolites would be expected to best capture the multidimensionality and interrelations of nutrients and foods present in dietary patterns. Several limitations of this study should be considered when interpreting the results. This study is observational by design. Although we have examined and adjusted confounders rigorously, the residual confounding cannot be completely ruled out. We assessed dietary intake at 8–13 GW and collected blood samples at 15–26 GW, with the median time interval (IQR) of 5.9 (4.4, 7.9) weeks. It is possible that some participants changed their diet during this interval. However, our dietary intake assessed by the FFQ covered the first trimester in which pregnancy caused nausea and vomiting occur. The influence of dietary change could be minor. Another limitation is that we have only included known metabolites in this study and thus some important metabolic features could possibly be missed.

## 5. Conclusions

In conclusion, we detected a set of metabolites associated with the DASH dietary pattern in pregnant women of multiple racial and ethnic groups. This is the first study to identify metabolomic biomarkers associated with the DASH dietary pattern among pregnant women. Several of the identified metabolites are also reported in previous studies conducted with general populations. Our study using the metabolomic approach contributes to the broader field of nutritional epidemiology by advancing our understanding of how DASH affects pregnant individuals at the molecular level. Identifying metabolomic markers associated with the DASH diet can help researchers and healthcare professionals understand how the body’s metabolism responds to this healthy dietary pattern. Metabolomic markers may also serve as biomarkers for adherence to the DASH diet. Monitoring these markers could help individuals and healthcare providers assess whether someone is following the diet and if it is having the desired impact on health outcomes. Future studies, in particular, intervention studies, are warranted to replicate our findings.

## Figures and Tables

**Figure 1 nutrients-16-00492-f001:**
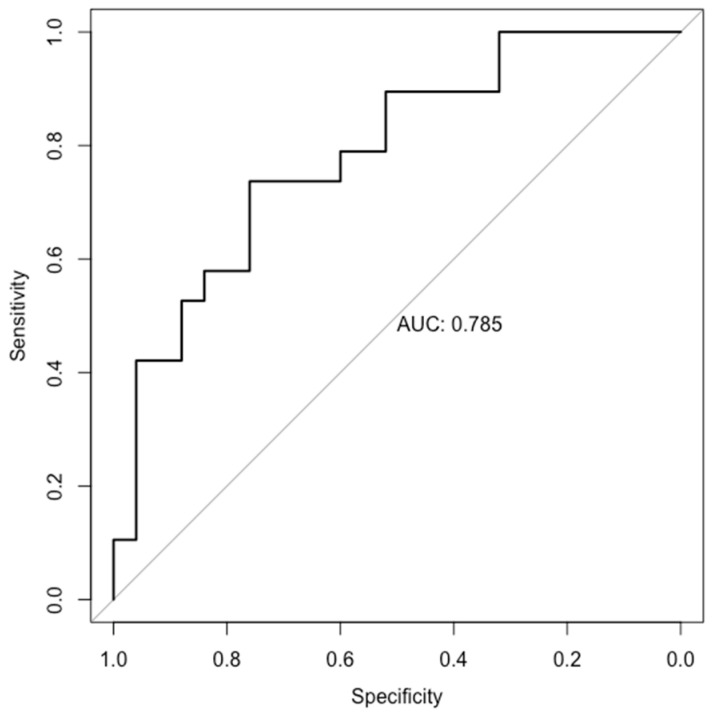
Receiver operating characteristic curve (ROC) for a 25-metbolite panel associated with DASH score in LASSO regressions with 10-fold cross-validation in the Fetal Growth Studies–Singletons cohort (FGS) study. The panel includes TG (49:1), TG (52:2), TG (54:5), PC (31:0), PC (35:3), PC (36:4) C, PC (36:5) B, PC (38:4) B, PC (p-42:4), PC (42:6), SM (d32:0), CE (18:2), gamma-tocopherol, hydroxycarbamate, glycine, xylitol, taurine, beta-alanine, asparagine, maleimide, Gal-Gal-Cer (d18:1/16:0), cystine, pyrophosphate, dodecanoic acid, and 3-aminoisobutyric acid.

**Table 1 nutrients-16-00492-t001:** Characteristics of pregnant women at enrollment (8–13 gestational weeks) from the NICHD Fetal Growth Studies–Singletons Cohort.

Characteristics	All (N = 186)	DASH Score Tertiles			
		T1 (N = 70)	T2 (N = 54)	T3 (N = 62)	*p*
*DASH score (median, IQR)*	*25 (22–28)*	*21 (19–22)*	*25 (24–26)*	*30 (28–31)*	
Age, years	28.00 (0.40)	26.95 (0.63)	28.72 (0.79)	28.46 (0.68)	0.15
Race/ethnicity, % (N)					<0.001
Non-Hispanic Whites	27.18 (37)	20.18 (10)	30.03 (12)	31.75 (15)	
Non-Hispanic Blacks	25.70 (23)	31.60 (12)	17.99 (5)	25.85 (6)	
Hispanics	24.39 (70)	26.84 (28)	27.28 (20)	19.86 (22)	
Asian/Pacific Islanders	22.73 (56)	21.38 (20)	24.70 (17)	22.54 (19)	
Education, N (%)					0.001
High-school or less	47.15 (84)	52.34 (38)	40.74 (20)	46.99 (26)	
Associates or higher	52.85 (102)	47.66 (32)	59.26 (34)	53.01 (36)	
Married/living with a partner, % (N)	70.14 (147)	76.41 (56)	64.79 (40)	68.17 (51)	<0.001
Born in the United States, % (N)	66.99 (99)	73.72 (39)	71.93 (29)	56.83 (31)	<0.001
Health insurance with private or managed care, % (N)	63.71 (123)	55.06 (41)	70.70 (36)	66.75 (46)	<0.001
Nulliparous, % (N)	50.54 (81)	35.79 (23)	59.37 (28)	58.03 (30)	<0.001
Consumed alcoholic beverage 3 months before pregnancy, % (N)	61.52 (113)	55.06 (42)	75.68 (35)	57.08 (36)	<0.001
Pre-pregnancy BMI, kg/m^2^	25.51 (0.38)	26.64 (0.68)	26.54 (0.77)	23.67 (0.43)	0.001
Pre-pregnancy BMI status, % (N)					<0.001
Normal (BMI < 25.0, kg/m^2^)	50.38 (84)	49.79 (29)	43.72 (21)	55.95 (34)	
Overweight (BMI: 25.0–29.9 kg/m^2^)	34.80 (63)	26.28 (23)	38.86 (21)	39.91 (19)	
Obese (BMI > 30, kg/m^2^)	14.82 (39)	23.93 (18)	17.42 (12)	4.14 (9)	
Total physical activity, metabolic equivalent hours/week	307.61 (10.59)	339.12 (20.03)	303.48 (19.88)	280.56 (14.18)	0.06
Dietary intakes ^§^,					
Total energy, kcal/day	2191.16 (71.19)	1876.97 (81.76)	2150.24 (135.83)	2522.49 (137.75)	<0.001
Total carbohydrate, g/day	299.60 (10.82)	244.91 (12.90)	275.49 (17.25)	367.32 (21.40)	<0.001
Total protein, g/day	86.15 (2.98)	73.58 (3.29)	82.96 (5.86)	100.58 (5.73)	<0.001
Total fatty acids, g/day	77.71 (2.81)	70.39 (2.94)	84.56 (6.23)	79.56 (5.36)	0.12
Saturated fatty acids (SAT), g/day	25.51 (1.03)	24.52 (1.16)	26.51 (2.29)	25.71 (1.98)	0.75
Monounsaturated fatty acids (MUFA), g/day	29.65 (1.13)	26.42 (1.13)	32.77 (2.49)	30.40 (2.17)	0.07
Polyunsaturated fatty acids (PUFAs), g/day	16.40 (0.60)	13.81 (0.57)	18.97 (1.29)	16.94 (0.97)	<0.001
Total dietary fiber, g/day	22.37 (0.84)	15.88 (0.81)	22.22 (1.46)	28.70 (1.56)	<0.001
Cholesterol, mg/day	286.53 (11.88)	312.13 (18.10)	278.15 (22.24)	268.33 (21.57)	0.27
Vegetables, serving/day	3.66 (0.19)	2.67 (0.23)	3.48 (0.37)	4.73 (0.36)	<0.001
Whole fruit, serving/day	6.39 (0.45)	3.93 (0.44)	4.63 (0.59)	10.06 (0.94)	<0.001
Whole grain, g/day	27.65 (1.42)	18.53 (1.80)	31.09 (2.92)	33.79 (2.35)	<0.001
Nuts and legumes, serving/day	0.67 (0.05)	0.32 (0.04)	0.78 (0.11)	0.93 (0.08)	<0.001
Red/processed meat, serving/day	0.47 (0.03)	0.55 (0.04)	0.45 (0.05)	0.42 (0.06)	0.14
Low-fat diary, serving/day	1.52 (0.14)	0.65 (0.16)	0.99 (0.14)	2.75 (0.30)	<0.001
Sodium, mg/d ^‡^	3598.52 (60.64)	3129.12 (122.93)	3573.03 (271.13)	3730.03 (229.77)	0.09
Sugar-sweetened beverages, serving/day	0.69 (0.09)	1.07 (0.16)	0.73 (0.21)	0.29 (0.08)	<0.001

Data are presented as weighted percentage and unweighted frequency % (N) for categorical variables and weighted mean (standard errors, SE) for continuous variables. IQR: interquartile range: 25th–75th percentiles. Sampling weights were applied to all analyses to represent the full NICHD Fetal Growth Studies–Singletons population. *p*-values were compared across three DASH score tertiles using one-way ANOVA for continuous variables and χ^2^ tests for categorical variables. ^§^ Dietary intakes were calculated among 186 women who completed the Food Frequency Questionnaires and had plausible total energy intake (i.e., 600–6000 kcal/day). ^‡^ Adjusted for total energy intake.

**Table 2 nutrients-16-00492-t002:** Eighteen fasting plasma metabolites that were both individually and jointly associated with DASH score in LASSO and linear regressions, the NICHD Fetal Growth Studies–Singletons cohort.

Superclass ^1^	Class/ Subclass	Metabolites	Direction of the Association in LASSO ^2^	Association in Linear Regression ^3^		Associated Food Groups (Regression Coefficients)
				*Coefficients*	*Benjamini–Hochberg-adjusted p-values ^3^*	
Organic acids and derivatives	Amino acids	Asparagine	Positive	0.13	0.03	Sodium (5 × 10^−5^)
		Beta-alanine	Positive	0.18	0.03	Low-fat dairy (0.04)
		Glycine	Negative	−0.18	0.003	Low-fat dairy (−0.05)
		Taurine	Negative	−0.49	0.003	/ ^4^
	Amino acid derivatives	Hydroxycarbamate	Positive	0.32	<0.001	Low-fat dairy (0.04)
Organoheterocyclic compounds	Pyrrolidines	Maleimide	Positive	0.25	0.04	/
Organic oxygen compounds	Carbohydrate/Monosaccharides	Xylitol	Positive	0.16	0.01	/
Lipids and lipids-like molecules	Fatty acyls	Dodecanoic acid	Positive	0.22	0.02	Fruit (0.02) and low-fat dairy (0.05)
	Prenol lipids/vitamin E	Gamma-tocopherol	Negative	−0.19	0.03	/
	Glycerolipids	TG (49:1)	Positive	0.38	<0.001	Low-fat dairy (0.07)
		TG (52:2)	Negative	−0.08	0.01	/
	Glycerophospholipids	PC (31:0)	Positive	0.25	0.002	Whole grains (0.01), low-fat dairy (0.07), and sodium (6 × 10^−5^)
		PC (35:3)	Positive	0.23	0.01	Low-fat dairy (0.07) and sodium (6 × 10^−5^)
		PC (36:4) C	Positive	0.07	0.001	Fruits (0.01)
		PC (36:5) B	Positive	0.43	0.002	/
		PC (38:4) B	Positive	0.16	<0.001	Low-fat dairy (0.04) and sodium (3 × 10^−5^)
		PC (42:6)	Positive	0.60	<0.001	Low-fat dairy (0.09)
	Sphingolipids	SM (d32:0)	Positive	0.29	0.003	Low-fat dairy (0.10) and sodium (8 × 10^−5^)

^1^ Classification of chemical compound classes was performed using ClassyFire. ^2^ Linear and LASSO regression models were adjusted for age (years), race (non-Hispanic White, non-Hispanic Black, Hispanic, and Asian and Pacific Islander), education (high-school degree or less, associated degree, and bachelor’s degree or more), pre-pregnancy BMI (kg/m^2^), and physical activity (metabolic equivalent hours per week). ^3^ Benjamini–Hochberg procedure was applied to adjust the multiple comparisons and statistical significance was considered for the false discovery rates (FDRs) <0.05. ^4^ “/”: no association with any foods. Abbreviations: BMI, body mass index; DASH, Dietary Approaches to Stop Hypertension; FDR, false discovery rate; PC, phosphatidylcholine; SM, sphingomyelin; TG, triacylglycerol.

## Data Availability

The data and codebook, along with a set of guidelines for researchers requesting the data, will be posted in the future to a data-sharing site, the NICHD/DIPHR Biospecimen Repository Access and Data Sharing (https://brads.nichd.nih.gov) (BRADS). The analytic code for this manuscript is available upon request.

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
