# Peer review of "Metabolomic Biomarkers of Dietary Approaches to Stop Hypertension (DASH) Dietary Patterns in Pregnant Women"

_nutrients, 2024, doi:10.3390/nu16040492_

Round 1

Reviewer 1 Report

Comments and Suggestions for Authors

The manuscript submitted is interesting, although it has limitations due to the study design. It seems that a far better way to find associations between metabolites and components of the DASH diet is to conduct an intervention study based on the implementation of the DASH diet and close dietary monitoring of this as in the study by Kim et al, Hypertension 2023. However, since this is the first study on matabolomic markers of the DASH diet to a group of pregnant women, it may be a good starting point for further research.

It seems that in an observational study, there may be many more confounding factors than the authors assumed.  My concern is the timing of the dietary assessment and blood sampling. If the dietary assessment took place between 8 and 13 GW (i.e., the diet was assessed in the preceding 3 months, and therefore before or at the very beginning of pregnancy), and the blood was drawn between 15 and 26 GW, this means that the diet was probably not assessed at the time the blood sample was drawn. It seems that the dietary assessment and blood draw should have been done simultaneously, or possibly the FFQ slightly after the blood draw, rather than the other way around.

What was the time interval between conducting FFQ and taking blood samples? This is very important because during pregnancy the diet is quite variable due to the occurrence of food complaints such as nausea and vomiting.  The evaluation occurred in the first trimester, where such complaints are common, and the blood test in the second trimester, where they usually subside, so the diet may have changed a lot. Was the incidence of nausea, vomiting and changes in appetite assessed? Please explain why the study was planned this way and address these concerns. 

The paper is well written, with clearly described methodology and well presented results. One minor comment concerns Table 2. With the TG relationship (52:2), the low fat diary is missing in the last column. Also, it should be clarified what the "/" sign in the third column means.

Author Response

The manuscript submitted is interesting, although it has limitations due to the study design. It seems that a far better way to find associations between metabolites and components of the DASH diet is to conduct an intervention study based on the implementation of the DASH diet and close dietary monitoring of this as in the study by Kim et al, Hypertension 2023. However, since this is the first study on matabolomic markers of the DASH diet to a group of pregnant women, it may be a good starting point for further research.

It seems that in an observational study, there may be many more confounding factors than the authors assumed.  My concern is the timing of the dietary assessment and blood sampling. If the dietary assessment took place between 8 and 13 GW (i.e., the diet was assessed in the preceding 3 months, and therefore before or at the very beginning of pregnancy), and the blood was drawn between 15 and 26 GW, this means that the diet was probably not assessed at the time the blood sample was drawn. It seems that the dietary assessment and blood draw should have been done simultaneously, or possibly the FFQ slightly after the blood draw, rather than the other way around.

Re: We thank the reviewer for this comment. We aimed to detect metabolomic markers of the DASH diet that reflect the pregnant women’s habitual dietary intakes. Thus, we intended to measure pregnant women’s habitual dietary intake using the FFQ in their early pregnancy. We believe the dietary assessment should be assessed before the blood sample collection. We agree with the reviewer the optimal time interval between the dietary assessment and blood sample collection should be narrower. We agree with the review an intervention study would be a better design to identify the biomarkers for dietary intakes, but an observational study with a prospective design like ours is appropriate to generate results and guide future studies.    

What was the time interval between conducting FFQ and taking blood samples? This is very important because during pregnancy the diet is quite variable due to the occurrence of food complaints such as nausea and vomiting.  The evaluation occurred in the first trimester, where such complaints are common, and the blood test in the second trimester, where they usually subside, so the diet may have changed a lot. Was the incidence of nausea, vomiting and changes in appetite assessed? Please explain why the study was planned this way and address these concerns. 

Re: We calculated the mean duration between FFQ and blood sample collection. The median (IQR) was 5.9 (4.4, 7.9) weeks. We agree with the review the diet during early pregnancy could be variable due to the vomiting. Unfortunately, we did not assess the symptoms of nausea or vomiting. However, our FFQ covers most of the first trimester in which the vomiting and nausea occurred.  Again, we aimed to detect metabolomic markers of the DASH diet that reflect the pregnant women’s habitual diets and we intended to measure pregnant women’s habitual dietary intake before the biospecimen collection to ensure the temporality. We addressed the reviewer’s concern in the limitation section of the revised manuscript.

The paper is well written, with clearly described methodology and well presented results. One minor comment concerns Table 2. With the TG relationship (52:2), the low fat diary is missing in the last column. Also, it should be clarified what the "/" sign in the third column means.

Re: We thank the reviewer for picking up the missing information in Table 2. There were no foods associated with TG(52:2) in our study. We have clarified that in the revised manuscript.

Reviewer 2 Report

Comments and Suggestions for Authors

The Authors of the revised manuscript entitled "Metabolomic biomarkers of Dietary Approaches to Stop Hypertension (DASH) dietary patterns in pregnant women" performed well designed observational study combinig metabolomic data with nutritional data. This approach is widely used in modern nutritional studies. The fact, that they decided to study pregnant women as study population is of utmost importance, as diet of mothers has great impact on the health status of their children also in adulthood. The methods which were used in the study were appropriate and the statistical analyzes of obtained data were aslo correct.

I have two concerns, from which one is rather important as it is ethical issue.

1). It appears that in this study the Autors used data obtained during NICHD project. Did they informed their patients to be included in another study? Did the patients give their written consent to be enrolled in this metabolome study? Was this study approved by the Bioethical Commitee? These issues should be addressed in the manuscript.

2). References should be included into the sentence eg. "...hypertension [2-5]." instead of "....hypertension. [2-5]"

Author Response

The Authors of the revised manuscript entitled "Metabolomic biomarkers of Dietary Approaches to Stop Hypertension (DASH) dietary patterns in pregnant women" performed well designed observational study combinig metabolomic data with nutritional data. This approach is widely used in modern nutritional studies. The fact, that they decided to study pregnant women as study population is of utmost importance, as diet of mothers has great impact on the health status of their children also in adulthood. The methods which were used in the study were appropriate and the statistical analyzes of obtained data were aslo correct.

I have two concerns, from which one is rather important as it is ethical issue.

1). It appears that in this study the Autors used data obtained during NICHD project. Did they informed their patients to be included in another study? Did the patients give their written consent to be enrolled in this metabolome study? Was this study approved by the Bioethical Commitee? These issues should be addressed in the manuscript.

Re: Yes, all participants were informed and agreed that their blood samples would be used in future studies including genetic, epigenetic, and biomarker investigations. IRB review and arrival were obtained. Please find the statement in the Methods part under the section of Study Population and Design.

“Institutional Review Board approval was obtained for all participating clinical sites, the data coordinating center, and NICHD.”

2). References should be included into the sentence eg. "...hypertension [2-5]." instead of "....hypertension. [2-5]"

Re: We thank the reviewer for picking up this and we have fixed that in the revised manuscript.

Reviewer 3 Report

Comments and Suggestions for Authors

The paper is clear, well written and interesting.

A few comments:

-the people with the highest DASH scores are also those who consume the most calories and move the least. Furthermore, counterintuitively, these people have the highest sodium intake. A comment on this would be helpful.

-did you consider the intake of fatty acids in the regression models?

-what are the practical implications of your findings?

Author Response

The paper is clear, well written and interesting.

A few comments:

-the people with the highest DASH scores are also those who consume the most calories and move the least. Furthermore, counterintuitively, these people have the highest sodium intake. A comment on this would be helpful.

Re: We thank the reviewer for picking up this observation. We found participants with the highest DASH score (i.e., 3rd Tertile) did have the highest caloric intake. Given the high correlation between total energy and sodium among the foods, individuals who had higher caloric intakes could also have higher high sodium consumption. In our study, the correlation coefficient between dietary calorie and sodium was 0.89. After adjusting for total caloric intake, from Tertile 1 to Tertile 3 of DASH score, the mean (SE) sodium intake was 3598.52 (60.64), 3634.17 (126.69), and 3235.02 (106.45) mg. We presented the calorie-adjusted sodium intake in Table 1 of the revised manuscript.

-did you consider the intake of fatty acids in the regression models?

Re: We did not control for the fatty acid intake in the regression model for 2 reasons. First, we are interested in the overall dietary pattern, DASH, instead of any particular nutrients. Second, we consider fatty acids as the potential mediators. Future studies may need to test the mediation pathways using the metabolites profiled through the targeted metabolomic approach.  

-what are the practical implications of your findings?

Re: We thank the reviewer for pointing out the importance of the implications of our study findings. We have added the following statement regarding the scientific and practical implications in the revised manuscript.

“Our study using the metabolomic approach contributes to the field of nutritional epidemiology by advancing our understanding of how DASH affects pregnant individuals at the molecular level. Identifying metabolomic markers associated with the DASH diet can help researchers and healthcare professionals understand how the body's metabolism responds to this healthy dietary pattern.  Metabolomic markers may also serve as biomarkers for adherence to the DASH diet. Monitoring these markers could help individuals and healthcare providers assess whether someone is following the diet and if it is having the desired impact on health outcomes.”

Round 2

Reviewer 1 Report

Comments and Suggestions for Authors

Thank you for the clarification. I believe that the manuscript should be accepted in its current form. 

Reviewer 2 Report

Comments and Suggestions for Authors

Accept in present form.

Reviewer 3 Report

Comments and Suggestions for Authors

I have no further suggestions